



# Evaluation of Native Earth System Model Output with ESMValTool v2.6.0

Manuel Schlund[1], Birgit Hassler[1], Axel Lauer[1], Bouwe Andela[2], Patrick Jöckel[1], Rémi Kazeroni[1], Saskia Loosveldt Tomas[3], Brian Medeiros[4], Valeriu Predoi[5], Stéphane Sénési[6], Jérôme Servonnat[7], Tobias Stacke[8], Javier Vegas-Regidor[3,a], Klaus Zimmermann[9], and Veronika Eyring[1,10]

[1]Deutsches Zentrum für Luft- und Raumfahrt (DLR), Institut für Physik der Atmosphäre, Oberpfaffenhofen, Germany
[2]Netherlands eScience Center (NLeSC), Amsterdam, the Netherlands
[3]Barcelona Supercomputing Center (BSC), 08034, Barcelona, Spain
[4]Climate and Global Dynamics Laboratory, National Center for Atmospheric Research, Boulder, Colorado, USA
[5]NCAS-CMS, University of Reading, Reading, UK
[6]Stéphane Sénési EIRL, Colomiers, France
[7]Laboratoire des Sciences du Climat et de l'Environnement, Gif sur Yvette, France
[8]Max Planck Institute for Meteorology, Hamburg, Germany
[9]Swedish Meteorological and Hydrological Institute (SMHI), Folkborgsvägen 17, 601 76 Norrköping, Sweden
[10]University of Bremen, Institute of Environmental Physics (IUP), Bremen, Germany
[a]now at: Nnergix Energy Management SL, Avenida Josep Tarradellas 80, 08029 Barcelona, Spain

**Correspondence:** Manuel Schlund (manuel.schlund@dlr.de)

**Abstract.** Earth system models (ESMs) are state-of-the-art climate models that allow numerical simulations of the past, present-day, and future climate. To extend our understanding of the Earth system and improve climate change projections, the complexity of ESMs heavily increased over the last decades. As a consequence, the amount and volume of data provided by ESMs has increased considerably. Innovative tools for a comprehensive model evaluation and analysis are required to assess the performance of these increasingly complex ESMs against observations or reanalyses. One of these tools is the Earth System Model Evaluation Tool (ESMValTool), a community diagnostic and performance metrics tool for the evaluation of ESMs. Input data for ESMValTool need to be formatted according to the CMOR (Climate Model Output Rewriter) standard, a process that is usually referred to as *CMORization*. While this is a quasi-standard for large model intercomparison projects like the Coupled Model Intercomparison Project (CMIP), this complicates the application of ESMValTool to non-CMOR-compliant climate model output.

In this paper, we describe an extension of ESMValTool introduced in v2.6.0 that allows seamless reading and processing native climate model output, i.e., raw output directly produced by the climate model. This is achieved by an extension of ESMValTool's preprocessing pipeline that performs a CMOR-like reformatting of the native model output during runtime. Thus, the rich collection of diagnostics provided by ESMValTool is now fully available for these models. For models that use unstructured grids, a further preprocessing step required to apply many common diagnostics is regridding to a regular latitude-longitude grid. Extensions to ESMValTool's regridding functions described here allow for more flexible interpolation schemes that can be used on unstructured grids. Currently, ESMValTool supports nearest-neighbor, bilinear, and first-order conservative regridding from unstructured grids to regular grids.





Example applications of this new native model support are the evaluation of new model setups against predecessor versions,

assessing of the performance of different simulations against observations, CMORization of native model data for contributions to model intercomparison projects, and monitoring of running climate model simulations. For the latter, new general-purpose diagnostics have been added to ESMValTool that are able to plot a wide range of variable types. Currently, five climate models are supported: CESM2 (experimental; will be fully available in ESMValTool v2.7.0), EC-Earth3, EMAC, ICON, and IPSL-CM6. As the framework for the CMOR-like reformatting of native model output described here is implemented in a general

way, support for other climate models can be easily added.

## 1  Introduction

Earth system models (ESMs) are state-of-the-art numerical climate models designed to improve our understanding of mechanisms and feedbacks in present-day climate and to project climate change for different future scenarios. Current climate models evolved steadily from relatively simple atmosphere-only models to today's complex ESMs participating in the latest (sixth)

phase of the Coupled Model Intercomparison Project (CMIP6; Eyring et al., 2016). Over the last decades, the complexity of these ESMs heavily increased with the inclusion of more and more detailed physical, biological, and chemical processes, but also with a steady increase in the models' spatial resolution. Continuous improvement and extension of the models was and is needed to represent key feedbacks that affect climate change. However, this increasing complexity is also a possible driver for an increase in inter-model spread of climate projections within the multi-model ensemble as the degrees of freedom in

the models increase. At the same time, high-resolution models are being developed with the ultimate aim of being able to explicitly resolve small-scale processes, including clouds and convection. More than ever, these developments require innovative and comprehensive model evaluation and analysis tools to assess the performance of these increasingly complex and high resolution models (Eyring et al., 2019).

One of these software tools is the Earth System Model Evaluation Tool (ESMValTool; Righi et al., 2020; Eyring et al., 2020;

Lauer et al., 2020; Weigel et al., 2021). ESMValTool is a community-developed, open-source software tool for evaluation and analysis of output from ESMs that allows for comparison of results from single or multiple models, either against predecessor versions or observations. A particular aim of ESMValTool is to raise the standards for model evaluation by providing well documented source code, scientific background documentation of the diagnostics and metrics included, as well as a detailed description of the technical infrastructure. All output created by the tool is assigned a provenance record that allows for

traceability of the results by providing information on input data used, processing steps, diagnostics applied, and software versions used. ESMValTool version 2, initially released in 2020, has been optimized for handling the large data-volume of the output from CMIP6 (Eyring et al., 2016) but can also be used to evaluate, analyze, or monitor simulations from individual models. The core functionalities of ESMValTool (referred to as *ESMValCore*; see Righi et al., 2020) are written in Python and take advantage of state-of-the-art computational libraries such as Iris (Met Office, 2010 - 2013) and methods such as

parallelization and out-of-core computation (Dask; Dask Development Team, 2016) to allow for efficient and user-friendly data processing. Common operations on the input data such as horizontal and vertical regridding, masking of missing values





across different data sets, or computation of multi-model statistics are centralized in a highly optimized preprocessor and available to all diagnostics.

Originally, ESMValTool has been designed and applied to process and analyze the output from CMIP models (e.g., Bock et al., 2020). For this, the model output has to be formatted according to the CMIP data request (e.g., https://clipc-services.ceda. ac.uk/dreq/index.html, last access: 1 August 2022) and the Climate and Forcast (CF) conventions (https://cfconventions.org/, last access: 1 August 2022) regarding variable names, metadata, and file format. Usually, this is done with the Climate Model Output Rewriter (CMOR; see https://cmor.llnl.gov/, last access: 1 August 2022) based on the CMOR tables (e.g., https:// github.com/PCMDI/cmip6-cmor-tables, last access: 1 August 2022). This process is usually referred to as *CMORization* and the reformatted data can be described as *CMORized*. While this has become a quasi-standard for large model intercomparison projects such as CMIP, this hampers application of ESMValTool during model development cycles or for monitoring of running model simulations as native model output (i.e., rawoutput directly produced by the climate model) typically does not follow the CMOR standard and thus would have to be CMORized in an additional step before running ESMValTool.

Here, we describe an extension of ESMValTool that has been introduced with v2.6.0 (Andela et al., 2022a) to read and process native model output from five different ESMs: CESM2 (experimental; will be fully available in ESMValTool v2.7.0), EC-Earth3, EMAC, ICON, and IPSL-CM6. The description of the technical implementation and workflow is intended to serve as a blueprint for implementing further support for other models so that ESMValTool can be used directly with their native output. This extension allows processing native model output by making the data compliant with the CMOR standard during runtime (referred to as *CMOR-like reformatting* hereafter). This enables the application of the rich collection of diagnostics provided by ESMValTool to these models. For example, this can be used to evaluate new model versions or parameterizations against older versions of the same model. At the same time, the model output can also be compared with observations, reanalyses, and/or other models such as the CMIP6 models without having to spend time and energy on the relatively complex CMORizations of the model output using external tools. This makes the integration of ESMValTool into model development cycles as well as the application of ESMValTool for monitoring of simulations significantly easier and more user-friendly.

This paper is structured as follows: Section 2 provides a technical description of the CMOR-like reformatting of native model output and a brief overview for the five currently supported models. Section 3 describes the currently available regridding functionalities for data on unstructured grids (grids defined by a list of latitude/longitude values) including an extension that allows a more flexible specification of interpolation schemes. Sections 4 and 5 present two examples of the evaluation of native model output representative for the wide range of diagnostics provided by ESMValTool: the near real-time monitoring of running climate model simulations and the evaluation of ESMs in a multi-model context, respectively. The paper closes with a summary and outlook in Section 6.





## 2 CMOR-like Reformatting of Native Model Output

### 2.1 General Implementation

The CMOR-like reformatting of native model output during runtime is implemented into ESMValTool as part of the prepro-
cessing chain. As illustrated by Figure 1, this preprocessing handled by the ESMValCore package (light blue box) is the first
of two main steps in ESMValTool's data flow that transforms the raw input data into preprocessed data. In the second main
step, these preprocessed data are then transformed into output (graphic, netCDF, and log files) applying diagnostics (orange
box). Within the preprocessor, the CMOR-like reformatting is implemented using model-specific automated *fixes* (highlighted
small yellow box). Usually, these fixes are used to correct minor errors in the input files such as invalid metadata or wrong
units (Righi et al., 2020). Here, we extend the functionalities of these fixes to reformat the native model output during runtime.

There are three different types of fixes (large yellow boxes at the top of Figure 1): (1) variable-specific fixes that are only
applied to a single variable of the native model output, (2) MIP (Model Intercomparison Project) table–specific fixes that are
applied to all variables of a specific table (e.g., Amon or Omon), and (3) model-specific fixes which are applied to all variables
of a specific model. Thus, when reading a specific variable with ESMValTool, up to three different fixes may be used. Usually,
the bulk of the CMOR-like reformatting procedure (mainly adding/modifying required coordinates and variable metadata) is
implemented in the model-specific fixes (3). If a variable is not directly available in the native model output but has to be derived
from other variables (e.g., total precipitation as the sum of large-scale precipitation, convective precipitation and snowfall), this
can be done in the variable-specific fixes (1). MIP table–specific fixes (2) are used to change/add metadata required for all
variables of a MIP table, e.g., to add a required scalar depth coordinate for ocean surface variables.

Each type of fix is implemented as a Python class with the name of this class determining its type. Note that this naming
convention also follows the PEP 8 style guidelines (https://peps.python.org/pep-0008/, last access: 1 August 2022); thus, all
class names are capitalized. The variable-specific fix classes (1) are named like the variable they are applied to (e.g., `Tas` for
the CMOR variable *tas*), the MIP table–specific fix classes (2) have the name of the corresponding MIP table (e.g., `Amon`
or `Omon`), and the model-specific fix class (3) is called `AllVars`. All of these classes need to be contained in a single file
(e.g., in the file `icon.py` for the CMOR-like reformatting of ICON). Each fix class can contain up to three fix functions that
are executed at different stages of the preprocessor: `fix_file`, `fix_metadata` and `fix_data`. As the very first step in
the preprocessing chain, `fix_file` is meant to fix input files that cannot be read by the ESMValTool preprocessor (via the
Iris module) without modifications. However, this step is only necessary in very rare cases. `fix_metadata` is designed to
fix metadata issues right after loading the input files with Iris. This function takes all variables of a file as an input. Finally,
`fix_data` is applied to data sets after extracting the desired time ranges from the input files and concatenating them into
a single object. This function takes only the desired variable as an input and contains potentially time-consuming fixes that
should not be applied to all input files but rather only to the subset of data requested by the user. However, in practice, most fixes
only use `fix_metadata` even when the actual data need to be modified. The reason for this is the different call signatures of
`fix_metadata` and `fix_data`: while `fix_metadata` takes all available variables of the input files as input, `fix_data`





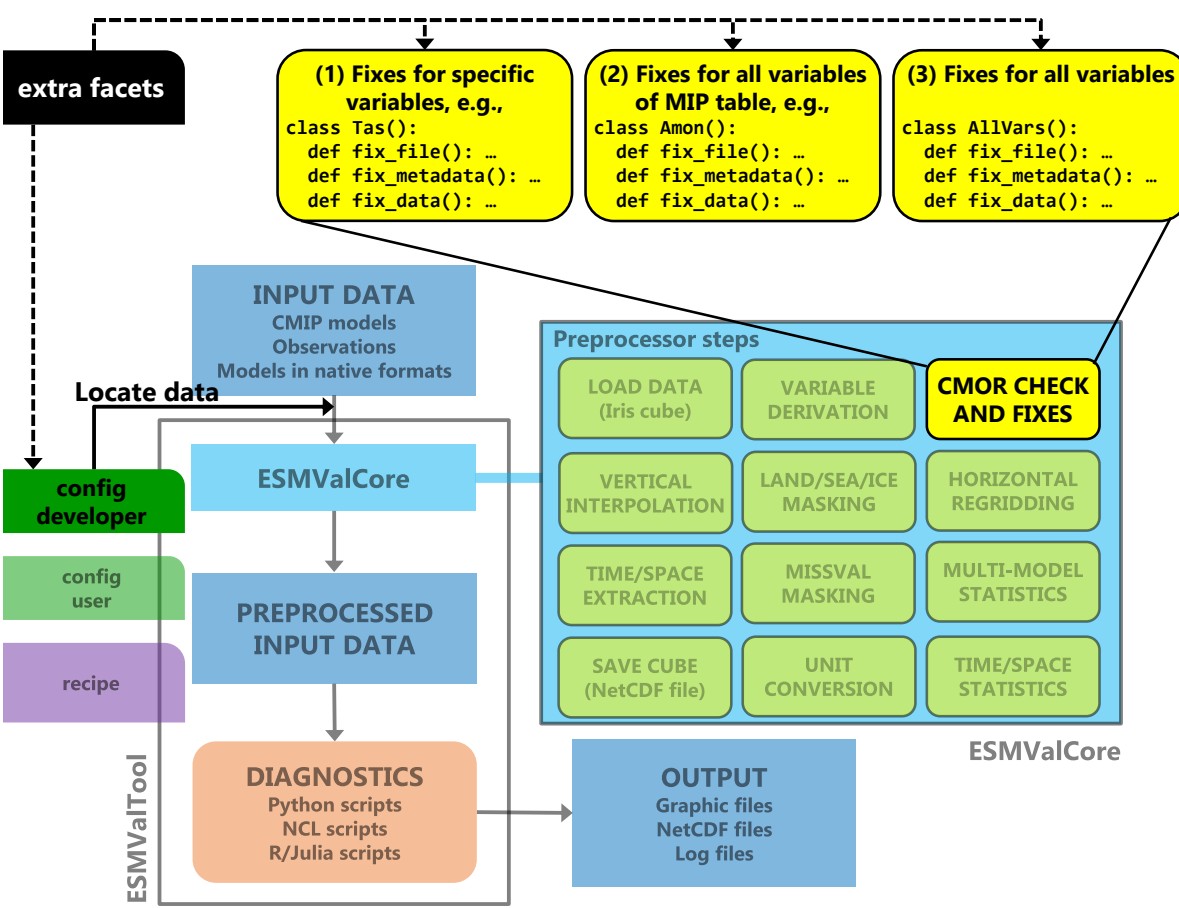

**Figure 1.** Schematic representation of ESMValTool. Non-transparent colors correspond to elements that have been altered/added to allow CMOR-like reformatting of native model output during runtime, which is included into ESMValTool as part of the preprocessing chain. Adapted from Righi et al. (2020).

only uses the single requested variable. An example where this is necessary is the variable derivation mentioned above, in which a CMOR variable is calculated from one or multiple other variables present in the input files.

ESMValTool expects a specific format for names of input files and directories (Data Reference Syntax, DRS; e.g., https://pcmdi.llnl.gov/mips/cmip5/docs/cmip5_data_reference_syntax.pdf, last access 1 August 2022). Default values for these naming conventions are specified in the file `config-developer.yml` (green box on the left in Figure 1). However, by using a

custom `config-developer.yml` file, arbitrary DRS formats for input files and directories can be considered. These input conventions can be configured separately for each supported *project*. In this context, a project refers to a model intercomparison project (e.g., CMIP3, CMIP5, CMIP6, etc.) or a type of observational product (e.g., OBS, obs4MIPs, etc.). However, since the structure and format of native model output can be very diverse, here *project* may also refer to the name of the model in its native format, e.g., `project: ICON` for the ICON model. Note that while for projects like CMIP6 or OBS the key





`dataset` refers to the name of the model or observational product, for native model output it refers to a sub version of the model or simply repeats the name from the project, e.g., `dataset: ICON` for the ICON model. Due to technical reasons, it is not possible to omit the key `dataset` although it may be redundant in some cases.

To facilitate the handling of native model output, ESMValTool now also allows the automatic addition of *extra facets* to the variable metadata (black box at the top of Figure 1). The term *facet* here refers to key-value pairs that describe data

sets requested by the user in an ESMValTool recipe, e.g., `project: CMIP6`, `dataset: CanESM5`, `mip: Amon`, `exp: historical`, or `short_name: tas`. These extra facets are automatically added to the original facets (if not already present) depending on the project, data set name, MIP table and variable requested by the user. By default, extra facets are read from a YAML file (https://yaml.org/, last access: 1 August 2022) contained in the ESMValTool repository. If needed, a custom location for this file can be specified by the user. An example of extra facets for the EMAC model is given in Appendix A. In

the context of reading native model output, extra facets can be used to locate input data. For example, if native model output is structured in subdirectories, the name of the corresponding subdirectory for each variable can be conveniently added through extra facets. This avoids the necessity to include this information in the ESMValTool recipe and the users do not need to be familiar with the peculiarities of each model. In addition, extra facets are also directly passed to the fix classes mentioned above. This can be used to further configure the fix operations applied to the data without alterations of the code.

## 2.2 Supported Models


Currently, ESMValTool supports the CMOR-like reformatting of native model output for five different models: CESM2 (experimental; will be fully available in ESMValTool v2.7.0), EC-Earth3, EMAC, ICON, and IPSL-CM6. The following subsections describe details on the implementations of these five reformatting procedures. All of them fix variable and coordinate metadata (names and units) not compliant with the CMOR standard and add missing scalar coordinates (e.g., 2m-height coordinate for

the near-surface air temperature) by default.

### 2.2.1 CESM2

CESM2 is an ESM developed by the National Center for Atmospheric Research (NCAR) in collaboration with a global community of users and developers (Danabasoglu et al., 2020). Like other ESMs, CESM2 is composed of several component: the Community Atmosphere Model, version 6 (CAM6); the Parallel Ocean Program Version 2 (POP2; Danabasoglu et al., 2012); the

Community Land Model, version 5 (CLM5; Lawrence et al., 2019); the Los Alamos sea ice model, version 5 (CICE5; Hunke et al., 2015); and the Model for Scale Adaptive River Transport (MOSART; Li et al., 2013). Additionally, CESM2 has the capability to represent the Greenland ice sheet using the Community Ice Sheet Model Version 2.1 (CISM2.1; Lipscomb et al., 2019) and the ocean biogeochemistry using the Marine Biogeochemistry library (MARBL; Long et al., 2021). The coupling between components is achieved through the Common Infrastructure for Modeling the Earth (CIME; http://github.com/ESMCI/cime,

last access 1 August 2022).

Output from CESM2 consists of netCDF files. Configuration of output variables, frequency, sampling (i.e., average, instantaneous, minimum, or maximum), and other aspects can be set by users via namelist files. The output files are time-





slice files consisting of the specified variables at the specified frequency. The most common use case is to put monthly averages of many variables into files, with one month per file. For CMIP6, the conversion of these native files to CMOR-compliant files was done with a custom tailored workflow based on Python 2 (see https://github.com/ncar/pyconform and https://github.com/NCAR/conform-input; last access 1 August 2022). In contrast to the other four models presented in this paper, ESMValTool's support for native CESM2 output is still experimental and under active development. It will be fully available in ESMValTool v2.7.0.

### 2.2.2   EC-Earth3

EC-Earth3 is a global climate model developed as part of a European consortium led by the Swedish Meteorological and Hydrological Institute SMHI (Döscher et al., 2022). The model is composed of several coupled components to describe the atmosphere, ocean, sea ice, land surface, dynamic vegetation, atmospheric composition, ocean biogeochemistry, and the Greenland Ice Sheet domains. Atmospheric and land dynamics are represented using the European Centre for Medium-Range Weather Forecast's (ECMWF) Integrated Forecast System (IFS) Cycle 36r4 (e.g., https://www.ecmwf.int/node/14597, last access: 1 August 2022), whereas the ocean is simulated using NEMO3.6 (Madec, 2008, 2015; Madec et al., 2017) which integrates LIM3 (Vancoppenolle et al., 2009; Rousset et al., 2015) and PISCES (Aumont et al., 2015) to represent the ocean biogeochemical and sea ice processes, respectively. Simulation of dynamic vegetation processes is performed by LPJ-GUESS (Smith et al., 2014; Lindeskog et al., 2013). Aerosols and chemical processes are described by TM5 (van Noije et al., 2014), and the Greenland Ice Sheet is modeled using PISM (Bueler and Brown, 2009; Winkelmann et al., 2011). The coupling of all components is performed using the OASIS3-MCT coupling library (Craig et al., 2017).

EC-Earth3 produces output in netCDF format for the ocean and the sea ice domains, and in GRIB format for the atmosphere and land domains. The reformatting and CMORization of the native model output to netCDF format that follows the CF conventions is performed as a step in the workflow running the simulations using the Python package ece2cmor3 (van den Oord, 2017, https://github.com/EC-Earth/ece2cmor3, last access: 1 August 2022) based on the CMOR standard, and which contains modules to format each of the model components. Thus, a CMOR-like reformatting of the native EC-Earth3 output within ESMValTool during runtime is not necessary. Nevertheless, ESMValTool includes several data and metadata fixes for EC-Earth3 to fully correct issues that have not been handled by ece2cmor3 to ensure consistency over experiments.

### 2.2.3   EMAC

The ECHAM/MESSy Atmospheric Chemistry (EMAC) model is a numerical chemistry and climate model system that includes submodels for tropospheric and middle atmosphere processes and their interactions with the ocean, land and human influences (Jöckel et al., 2010). It uses the second version of the Modular Earth Submodel System (MESSy2) to link multi-institutional computer codes. The core atmospheric model is the 5th generation European Centre Hamburg general circulation model (ECHAM5; Roeckner et al., 2006). The physics subroutines of the original ECHAM code have been modularized and reimplemented as MESSy submodels and have been continuously further developed. Only the spectral transform core, the flux-form semi-Lagrangian large scale advection scheme (Lin and Rood, 1996), and the nudging routines for Newtonian relax-





ation are remaining from ECHAM. In MESSy, the memory, data types, metadata and output is handled by the infrastructure submodel CHANNEL (Jöckel et al., 2010), which allows a flexible control of the model output via two Fortran namelists. This includes output redirection to create custom tailored output files, the choice of the output file format, of the output method (e.g. serial vs. parallel netCDF), of the output precision, of the output frequency, and the capability to conduct basic statistical analyses w.r.t. time during runtime, i.e., to output in addition (or alternative) to the instantaneous data (i.e., at a specific model time step) the time average, standard deviation, minimum, maximum, event counts, and event averages for the output time interval.

To reformat EMAC, many variable-specific fixes are required since a large number of CMOR-type variables are not directly present in the native model output but need to be derived from other variables. For example, the variable *pr* (total precipitation) is calculated as the sum of the large-scale precipitation, convective precipitation, and snow fall. Consequently, a rather large amount of information needs to be provided in the form of extra facets. This includes raw variable names used in EMAC output files (only necessary if they differ from their corresponding CMOR variable names) and information on the EMAC *channel*. The latter refers to the additional hierarchy used to structure the EMAC output (see above): the variables are stored in different channels; for each channel individual output files are provided by the model. The channel information given by the extra facets file serves as a default value. If a different channel is requested this can be specified in the ESMValTool recipe.

### 2.2.4 ICON

The ICON (ICOsahedral Non-hydrostatic) modeling framework, developed by the Max Planck Institute for Meteorology (MPI-M), the German Weather Service/Deutscher Wetterdienst (DWD), and partners, provides a unified modeling system for global numerical weather prediction (NWP) and climate modeling (Zängl et al., 2014). The CMOR-like reformatting of ICON output implemented in ESMValTool primarily targets evaluation of climate model simulations, but could be extended to NWP simulations in the future. The reformatting has been successfully tested with output from atmosphere-only simulations (ICON-A; Giorgetta et al., 2018) and fully-coupled ESM simulations (ICON-ESM, also known as ICON-Ruby; Jungclaus et al., 2022).

ICON model output already provides many CMOR variables in the correct form. Thus, very little variable-specific fixes and additional information in the form of extra facets is required. These extra facets can include raw variable names given in the ICON output files (only necessary if they differ from their corresponding CMOR variable names) and alternative names for the latitude and longitude coordinates (currently only affects the grid cell areas *areacella* and *areacello* as these are extracted directly from the ICON grid file).

As shown in Figure 2a, native ICON model output uses an unstructured grid whose triangular grid cells are derived from a spherical icosahedron by repeated subdivision of the spherical triangular cells into smaller cells (Giorgetta et al., 2018; Wan et al., 2013). Consequently, the CMOR-like reformatting of ICON requires fixing the spatial coordinate that describes this unstructured grid in addition to the latitude and longitude coordinates. If the grid information (latitude and longitude coordinates) is missing in an input file, which can be the case for ICON output depending on the model settings, it is automatically added during the CMOR-like reformatting using the corresponding grid file. This grid file is specified in the global netCDF attributes of the ICON file and is automatically downloaded if necessary. For the vertical grid, the ICON reformatting supports the terrain





following hybrid sigma height coordinates that are used by the ICON model (Giorgetta et al., 2018), but also a regular height coordinate that simply describes the altitude of the grid cells. If available in the input file, pressure levels (including bounds) are added to the ESMValTool output files.

To be able to compare native ICON output directly with other models, observational products, or reanalysis data, an additional preprocessing step is usually necessary to interpolate the ICON data to a regular grid. This can be done with ES-
MValTool's regridding preprocessor, which is described in detail in Section 3. However, ICON data can also be regridded by external tools like CDO (Climate Data Operators; Schulzweida, 2021) if needed by the user, since the CMOR-like reformatting also supports ICON data on regular grids. For example, if users require a regridding algorithm available in CDO but currently not supported by ESMValTool, the native model data can be regridded using CDO in an additional postprocessing step after running the model before being processed by ESMValTool.

### 235 2.2.5 IPSL-CM6

IPSL-CM6A-LR (herafter IPSL-CM6) is an ESM developed by the Institut Pierre-Simon Laplace Climate Modeling Center. It is composed of the LMDZ atmospheric model version 6A-LR (Hourdin et al., 2020), the ORCHIDEE land surface model (Krinner et al., 2005) version 2.0 and the NEMO oceanic model (Madec, 2008, 2015). The latter is based on version 3.6 stable of NEMO, which includes three major components: the ocean physics model NEMO-OPA (Madec et al., 2017), the sea ice
dynamics and thermodynamics model LIM3 (Vancoppenolle et al., 2009; Rousset et al., 2015) and the ocean biogeochemistry model PISCES (Aumont et al., 2015).

IPSL-CM6 uses the XIOS input/output system (Meurdesoif, 2017) which, combined with dr2xml (https://github.com/rigoudyg/dr2xml, last access: 1 August 2022), allows production of CMOR-compliant output directly at run time. However, this feature is not yet standard for IPSL-CM6 runs and activated only for simulations contributing to some MIPs. Typically,
simulations for IPSL-CM6 development use the native model output format which exists in two versions: *Output* and *Analyse*. The *Output* format consists of netCDF files which include output for a fixed-length period of time (usually one month) and for a group of variables (e.g., all atmospheric 3D variables). These files are grouped in directories that contain all periods for one (or more) variable groups. The *Analyse* format has been introduced to facilitate the analysis of the model: output files in this format include only one variable for a longer time period (up to the entire simulation period). The *Analyse* format can be
requested in addition to the *Output* format during setup of the model experiment.

Data in native format are not CMOR-compliant. However, since the files comply with other conventions like CF, only a small number of ESMValTool fixes is necessary for the CMOR-like reformatting of the data. Apart from common fixes that are applied to all native model data (adapting variable and coordinate metadata and the addition of scalar coordinates), a fix for an auxiliary time coordinate that is not CMOR-compliant needs to be applied. Extra facets for IPSL-CM6 include raw
variable names used in the native IPSL-CM6 output and information about the variable groups and directories used to store the corresponding variables.





## 3  Regridding Data on Unstructured Grids

Many state-of-the-art ESMs do not use rectilinear or curvilinear horizontal grids for the spatial discretization but unstructured grids instead. Unstructured grids are usually described by a list of all grid cells using a single spatial dimension. For each

grid cell in this list, latitude and longitude values for the central points (representative for the cell *face*) and bounds (cell *nodes*) are specified by additional variables. Grid cells of unstructured grids usually consist of polygons whose number of vertices is different than four. For example, the ICON model (see Section 2.2.4) uses triangular grid cells. Unstructured grids offer numerical advantages in terms of scalability and computational efficiency, and also often offer a more straightforward implementation of multi-resolution modeling (e.g., nested high-resolution grids in regions of interest).

However, the evaluation of native model output on unstructured grids is challenging: for example, the output of most observations or reanalyses is given on different (regular) grids (which complicates a direct comparison) and most ESMValTool diagnostics therefore expect data on regular grids. For this reason, a regridding preprocessor that is able to interpolate unstructured grids to regular grids is often crucial for evaluation of such native model output. Currently, ESMValTool provides three different regridding schemes that allow regridding from unstructured grids to regular grids: nearest-neighbor, bilinear, and first-

order conservative interpolation. While the first scheme supports unstructured data in arbitrary format (the only prerequisite is the existence of latitude and longitude coordinates), the latter two can only be used with data that follow the UGRID (Unstructured Grid) conventions (https://ugrid-conventions.github.io/ugrid-conventions/, last access: 1 August 2022). UGRID provides a systematic description of the topology of unstructured grids (e.g., it clearly defines the connectivity between the cell faces and nodes), which is necessary to perform the more complex regridding operations. Nearest-neighbor interpolation is natively

supported by Iris used in the ESMValTool preprocessor. Bilinear and first-order conservative regridding are supported via the iris-esmf-regrid package (https://github.com/SciTools-incubator/iris-esmf-regrid, last access: 1 August 2022), which collects and provides the Earth System Modeling Framework (ESMF; https://earthsystemmodeling.org/regrid/, last access: 1 August 2022) regridding schemes for Iris. The use of iris-esmf-regrid is possible due to an extension of ESMValTool's regridding functionalities that allows the usage of external regridding packages (in addition to native Iris schemes) with arbitrary options.

An example of regridding ICON data on an unstructured grid is illustrated by Figure 2. The left panel (a) shows the triangular grid cells of the native model output on an R2B4 grid with a horizontal resolution of about 160 km. The right panel (b) shows the data interpolated on a regular 2°x2° grid that has been regridded using ESMValTool's nearest-neighbor scheme. From a visual inspection, both fields are very similar. As an additional sanity check, we calculated the global mean near-surface air temperature for both grids, which gives almost identical values of 287.14K and 287.16K for the native grid and the interpolated

grid, respectively. Since native ICON output does not follow the UGRID conventions, only the nearest-neighbor scheme is currently supported for this model. However, in ESMValTool v2.7.0, the CMOR-like reformatting of ICON will include a first implementation to make ICON output fully UGRID-compliant during runtime of ESMValTool. First tests have shown promising results: the adapted ICON data could be successfully regridded with the first-order conservative algorithm provided by iris-esmf-regrid.



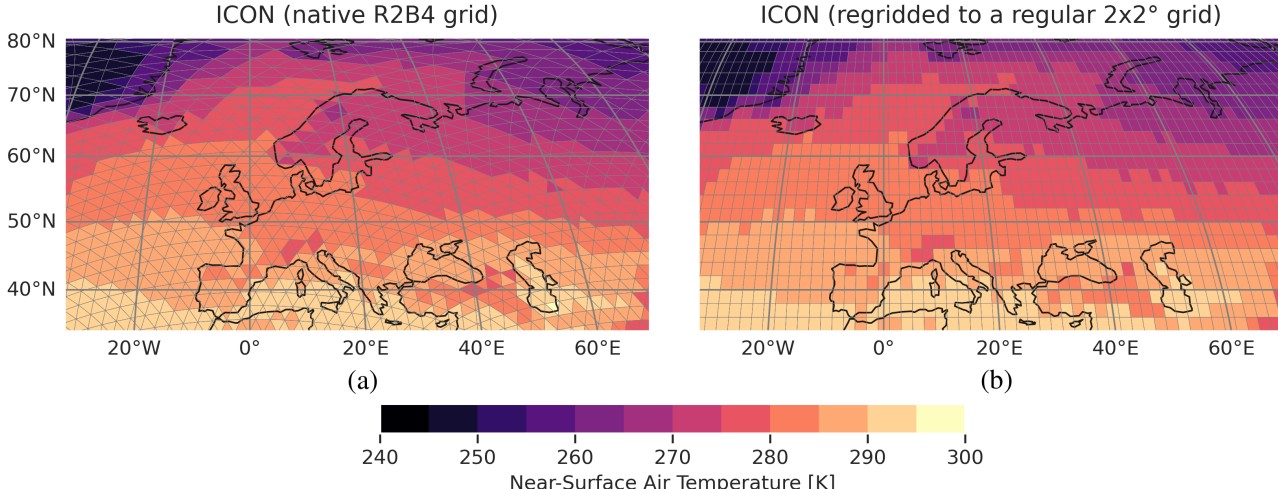

**Figure 2.** Illustration of the regridding of an unstructured grid using the near-surface air temperature climatology over Europe averaged from 1979 to 2014 as an example. The ICON simulation shown here corresponds to the one described in Figure 3. (a) Native ICON grid at R2B4 resolution (about 160 km). (b) Regular 2°x2° grid that results from ESMValTool's nearest-neighbor regridding of the data shown in (a).

We emphasize that regridding is not a trivial operation in general. ESMValTool's three currently available schemes for unstructured grids are sufficient for many applications; however, this is by no means a complete set of all possible regridding algorithms and does not cover all imaginable applications. For example, variables that describe fractions of quantities within grid cells like land/sea fraction, sea ice concentration, or fractional cloud cover need to be treated with extra care (e.g., Grundner et al., 2021). The nearest-neighbor scheme illustrated in Figure 2 is sufficient for the purpose of monitoring (i.e., to get a quick overview of simulation results), but should not be used for more sophisticated scientific analyses where precise results are crucial.

## 4 Monitoring of Running Climate Model Simulations

One use case of ESMValTool's new capability to process native model output is the near real-time monitoring of running climate model simulations. With this, modeling centers can already check at an early stage whether the output of their simulation appears to be reasonable. Possible problems can be detected very early on, which in turn can save valuable computational resources on supercomputers.

For the purpose of monitoring, a set of general diagnostics has been added to ESMValTool (see Table 1 for an overview). These diagnostics can be found in the subdirectory `diag_scripts/monitor`. All of these diagnostics are able to handle arbitrary variables from arbitrary data sets, which makes them versatile and flexible to use. The input for each diagnostic consists of data that have been preprocessed with ESMValTool. In order to configure the output, a number of parameters can be set and customized in the ESMValTool recipe that runs the diagnostic script. Settings related to the definition of the output





directies and filenames can also be configured in the ESMValTool recipe in order to store all output figures in a common location for each simulation following a common naming scheme. Furthermore, the path to an additional configuration file for the plots is also provided in the ESMValTool recipe. This configuration file contains map-specific settings for the map

plots (e.g., the map projection) and variable-specific settings (e.g., regions, titles, labels, and color schemes). Currently, this additional configuration file is only used by the diagnostic `monitor.py`.

The general purpose diagnostics are written in Python following an object-oriented implementation in order to facilitate the extension and inclusion of further monitoring diagnostics. To illustrate this procedure, the script `compute_eofs.py` has been developed following the same structure defined in the main `monitor.py` script. Since the monitoring diagnostics save

their output according to a customized but structured naming convention, the plot files can be easily used by other applications e.g. for visualization. For instance, in the case of monitoring EC-Earth3, an R Shiny app has been developed in order to conveniently and interactively visualize results by experiment, realm and variable. A screenshot of this application is shown in Figure B1. The following paragraphs illustrate five examples (one plot for each currently supported climate model) created

**Table 1.** Overview of the general-purpose monitoring diagnostics implemented in ESMValTool. All diagnostics can handle arbitrary variables from arbitrary data sets.

| Diagnostic (located in diag_scripts/monitor) | Brief description | Available plot types [+ example figure if present in this paper] |
|---|---|---|
| `monitor.py` | Basic plots to monitor running climate model simulations. Creates individual plots for each data set given in the ESMValTool recipe. | – Time series<br>– Annual cycles [see Figure 4]<br>– Maps (full climatologies, seasonal climatologies, and monthly climatologies) [see Figure 7] |
| `compute_eofs.py` | Calculate and plot empirical orthogonal functions (EOFs). Creates individual plots for each data set given in the ESMValTool recipe. | – Maps (EOFs)<br>– Time series (principal components) |
| `multi_datasets.py` | Combine multiple data sets in single plots. One input data set can be defined as reference, which will be used to plot biases. | – Time series [see Figure 3]<br>– Maps [see Figure 6]<br>– Profiles [see Figure 5] |

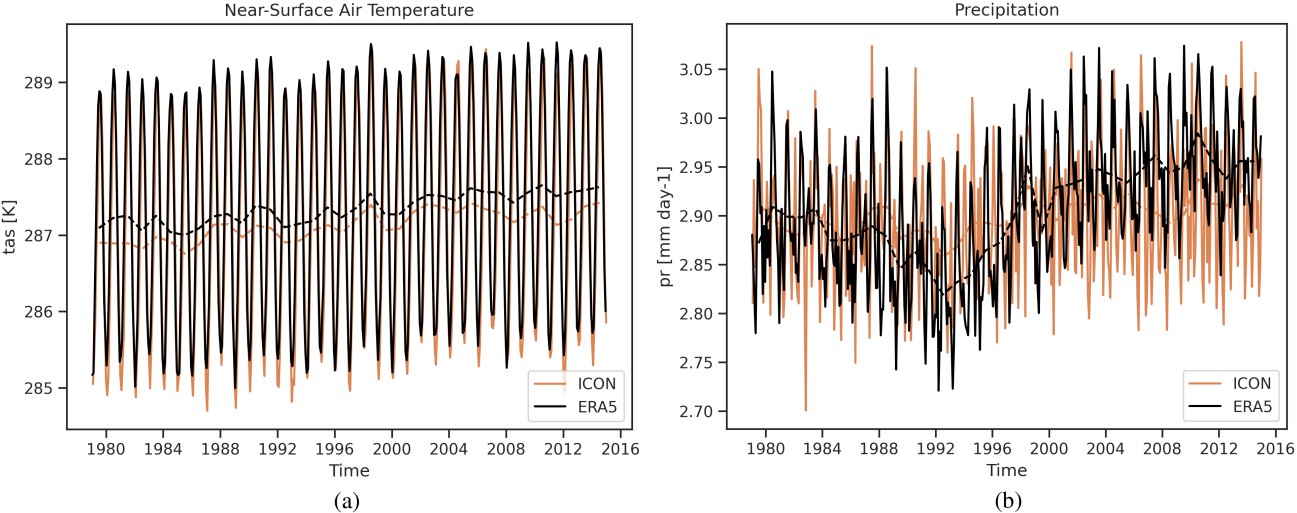

**Figure 3.** Monthly mean (solid lines) and annual mean (dashed lines) time series of ICON-ESM (orange) and ERA5 (black) for the period 1979 to 2014. The ICON simulation shown here (*Cool Ruby*) is based on a standard AMIP setup at R2B4 resolution (about 160 km) with an advanced representation of soil physics and properties. (a) Global mean near-surface air temperature. (b) Global mean precipitation.

with these new diagnostics. Please note that these figures only serve as examples and by no means represent the complete set
of available plots.

For a direct comparison with one or multiple reference data sets (e.g., observations or reanalyses), Figure 3 shows simple time series of the global mean near-surface air temperature and precipitation from 1979 to 2014 created by the diagnostic `multi_datasets.py` for the ESM configuration of ICON (ICON-ESM) and the ERA5 reanalysis (Hersbach et al., 2020). The ICON simulation shown here is conducted using a standard Atmospheric Model Intercomparison Project (AMIP) setup at
R2B4 resolution (about 160 km). In the CMIP terminology, the AMIP protocol refers to a simulation of the recent past with all natural and anthropogenic forcings, and prescribed sea surface temperatures and sea ice concentrations (Eyring et al., 2016). Compared to the standard ICON-ESM setup, this ICON version shown here (*Cool Ruby*) features an advanced representation of soil physics and soil properties. This plot type illustrated here is particularly suited to get a quick overview of climate model output and can be used early on in a simulation.

Apart from such time series, the monitoring diagnostics can also be used to visualize annual cycles of arbitrary variables. This plot type can be created with the diagnostic `monitor.py`. Figure 4 shows an example of this using the annual cycle of the global mean near-surface air temperature from CESM2. The simulation shown here also uses a standard AMIP setup as defined by CMIP6 with all forcings (anthropogenic and natural) from the recent past, and prescribed sea surface temperatures and sea ice concentrations.

In addition to the time series shown in Figure 3, the diagnostic `multi_datasets.py` also provides vertical profiles for a model and a reference data set including the difference between the two. If no reference data set is provided, a single vertical



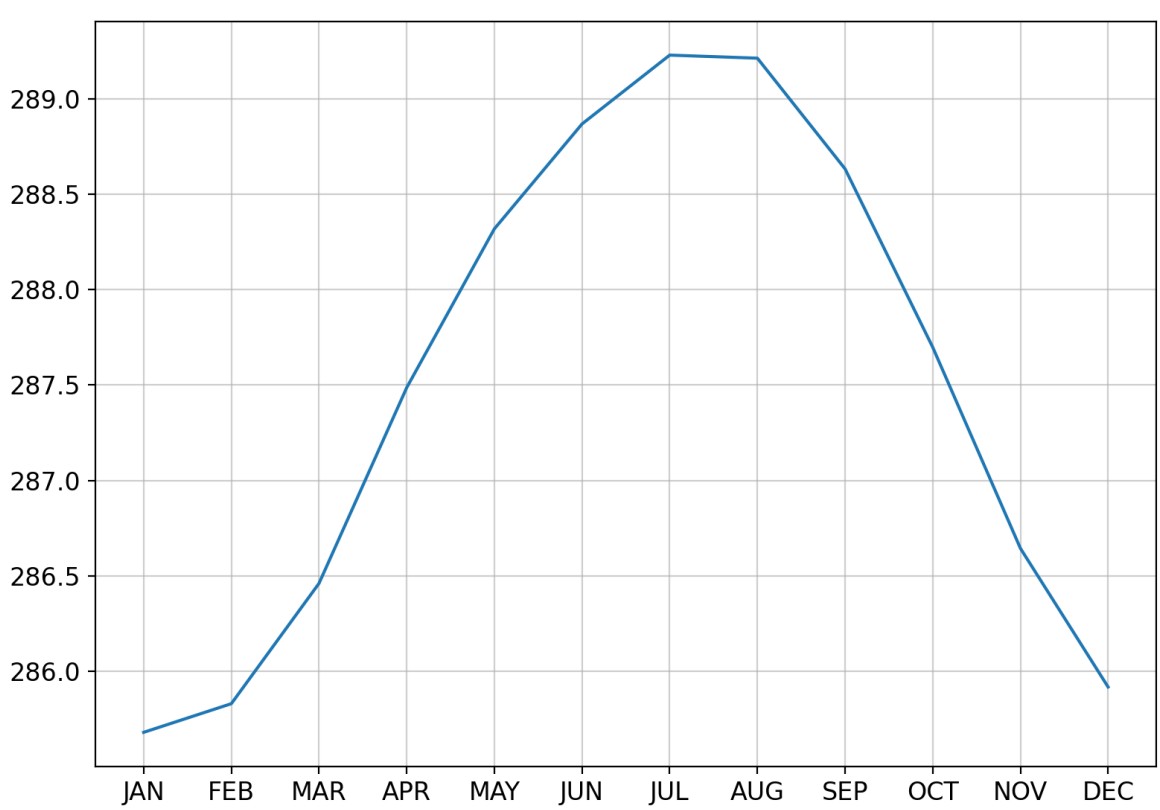

**Figure 4.** Annual cycle of the global mean near-surface air temperature from CESM2 averaged from 2005 to 2014. The CESM2 simulation shown here uses a standard AMIP setup with all forcings from the recent past and prescribed sea surface temperatures and sea ice concentrations.

profile of the model is returned. Figure 5 shows an example of the vertical air temperature profile from EMAC averaged over the years 2005 through 2014. These EMAC results are from the *RC2-base-04* simulation, which is a free running simulation following the Chemistry-Climate Model Initiative (CCMI-1) protocol (Jöckel et al., 2020). For details about the model setup

we refer to Jöckel et al. (2016). As reference data set, the ERA5 reanalysis is used here. The top row in the figure shows the vertical profile from EMAC (left) and ERA5 (right), while the bottom row shows the bias (calculated as simple difference) between the two data sets.

Moreover, `multi_datasets.py` also supports map plots (climatologies). Just like the vertical profiles provided by this diagnostic, these map plots can also be used to visualize differences between model data and a reference data set. As an



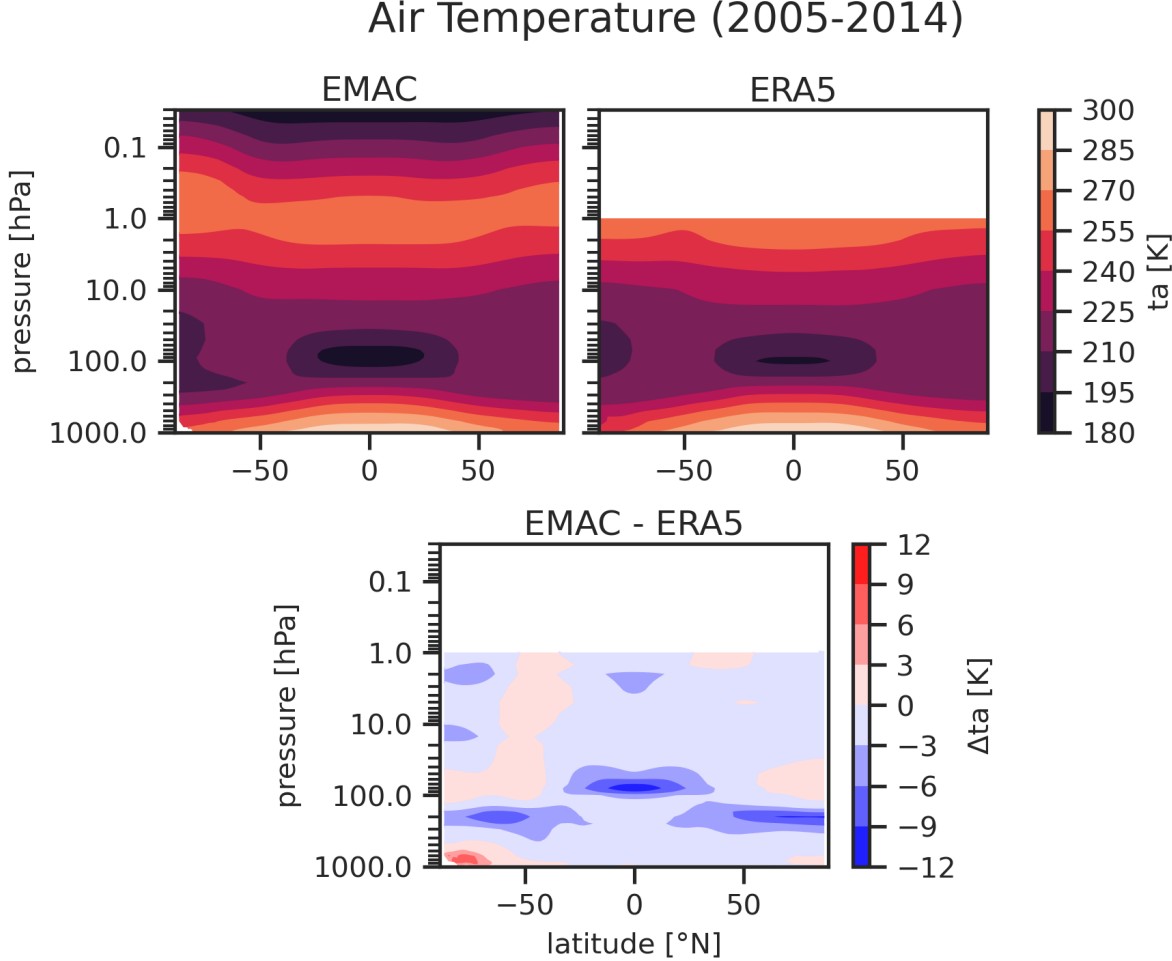

**Figure 5.** Zonal mean air temperature from EMAC including the bias relative to ERA5 averaged from 2005 to 2014. The EMAC results are from the *RC2-base-04* simulation (Jöckel et al., 2020), which is a free running simulation following the CCMI-1 protocol (see Jöckel et al. (2016) for details).

example, Figure 6 shows the global precipitation climatology from EC-Earth3-CC averaged over the years 2005 to 2014 in comparison to the ERA5 reanalysis. The panels are arranged similar to Figure 5: the top row shows the climatologies of EC-Earth3-CC (left) and ERA5 (right), the bottom row the difference between the two. The EC-Earth3-CC simulation shown is an AMIP simulation that has been published as part of the CMIP6 ensemble.

In contrast to the annual mean climatology given in Figure 6, Figure 7 shows monthly climatologies of the Arctic sea ice concentration for the months March and September averaged over the years 2005 to 2014 as simulated by IPSL-CM6. The simulation shown here is a member of the CMIP6 AMIP ensemble in its native format. This plot has been created with

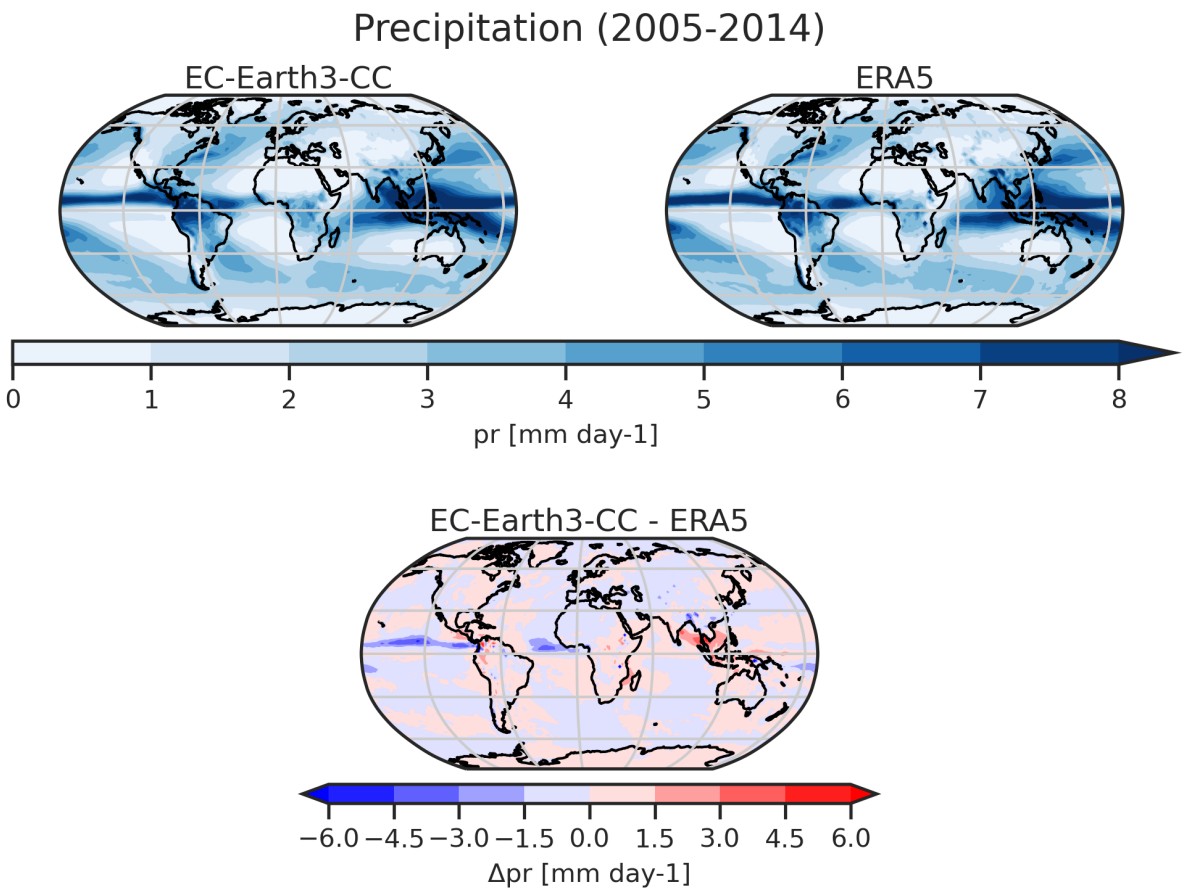

**Figure 6.** Global precipitation climatology from EC-Earth3-CC including the bias relative compared to ERA5 averaged over 2005 to 2014. The simulation shown here is an AMIP simulation that has been published as part of the CMIP6 ensemble.

`monitor.py`, which supports arbitrary regions and map projections. For example, here, a stereographic projection is used to focus on the Arctic region.

As mentioned above, the monitoring diagnostics provide further plot types which are not shown here. This includes (op-
tionally smoothed) time series and seasonal climatologies provided by the diagnostic `monitor.py`, and empirical orthogonal function (EOF) maps and time series provided by the diagnostic `compute_eofs.py`.

## 5 Availability of ESMValTool's Rich Set of Diagnostics for Native Model Output

The monitoring functionality described in the previous section of this paper is one possible application of ESMValTool's CMOR-like reformatting of native model output. In principle, the rich collection of diagnostics provided by ESMValTool
(see orange box in Figure 1) is now fully available for all supported models. This includes all diagnostics described in the



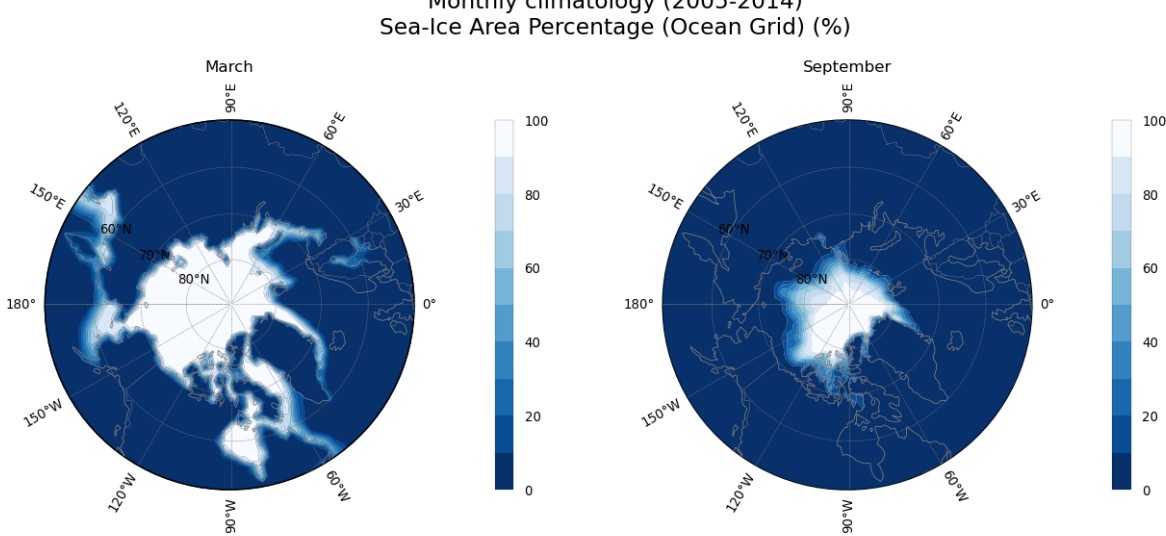

**Figure 7.** March and September Arctic sea ice concentration from IPSL-CM6 averaged over 2005 to 2014. The simulation shown here is a member of the CMIP6 AMIP ensemble in its native format.

scientific documentation of ESMValTool, e.g., large-scale diagnostics for a comprehensive evaluation of ESMs (Eyring et al., 2020), diagnostics for emergent constraints and future projections (Lauer et al., 2020), and diagnostics for extreme events, regional and impact evaluation (Weigel et al., 2021). Moreover, many new diagnostics have been added or will be added to ESMValTool, for example, diagnostics and recipes that have been used to compile parts of the latest Assessment Report 6

(AR6) of the Intergovernmental Panel on Climate Change (IPCC; e.g., Eyring et al., 2021).

As an example, Figure 8 shows the annual mean near-surface air temperature between 1979 and 2014 averaged over the tropical land (30°S–30°N) from the five models described in this paper that have been processed in their native format and an ensemble of (CMORized) CMIP6 models and the ERA5 reanalysis. A similar version of this plot has originally been published by Bock et al. (2020) to evaluate progress across different CMIP generations (CMIP3, CMIP5, and CMIP6). All data sets show

the steady increase of the near-surface air temperature over the last decades. For all CMIP6 models and the native output of the models CESM2, EC-Earth3-CC, ICON, and IPSL-CM6, this figure shows results of AMIP experiments. The native EMAC output shown here is from a free running EMAC simulation following the CCMI-1 protocol that also uses an AMIP-like setup with a different set of forcings (Jöckel et al., 2016). Figure 8 is just an example and we would like to note, that a fair comparison between the different results shown here is not possible because of the different model setups used. The main aim of this figure

is to showcase the evaluation of native model output alongside CMIP data and reanalysis products with ESMValTool's large collection of diagnostics.

The diagnostics presented in Sections 4 and 5 showcase two example application possible with ESMValTool v2.6.0. Further applications are, for example, comparison of newly developed model versions or setups with predecessor versions or



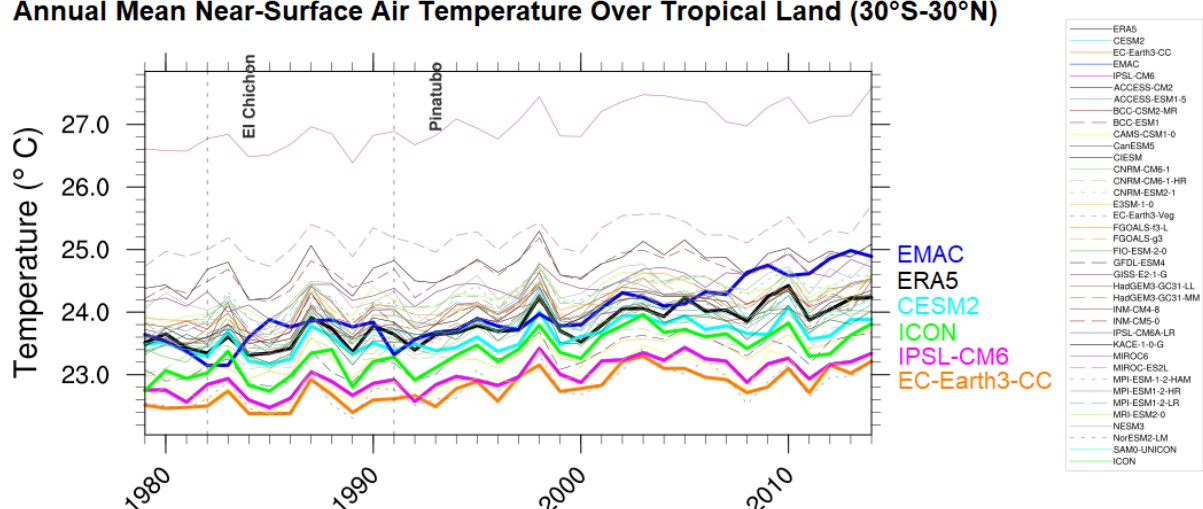

**Figure 8.** Example of an analysis of native model output alongside CMIP data and reanalysis products with ESMValTool's wide range of diagnostics, similar to Figure 1 of Bock et al. (2020): Annual near-surface air temperature between 1979 and 2014 averaged over the tropical land (30°S–30°N) for an ensemble of (CMORized) CMIP6 models (thin lines), the ERA5 reanalysis (thick black line), and the models presented in this paper for which a CMOR-like reformatting is available (CESM2: thick cyan line; EMAC: thick blue line; ICON: thick green line; IPSL-CM6: thick magenta line; EC-Earth3-CC: thick orange line). Vertical dashed lines show large volcanic eruptions. For all CMIP6 models and the native output of the models CESM2, EC-Earth3-CC, ICON, and IPSL-CM6, results of an AMIP simulation as defined by CMIP6 (Eyring et al., 2016) are used. The EMAC results shown here are based on a free running EMAC simulation following the CCMI-1 protocol that also uses prescribed sea surface temperatures and sea ice concentrations but a different set of forcings (Jöckel et al., 2016). Due to different model setups, a fair comparison of the individual models is not possible.

observations, or the plain CMORization of native model output prior to publication of the data as a contribution to model
intercomparison projects like CMIP.

## 6   Summary and Outlook

We have described recent changes and additions to ESMValTool that allow reading and processing of native model output through an automatic CMOR-like reformatting during runtime for five different climate models: CESM2, EC-Earth3, EMAC, ICON, and IPSL-CM6. Prior to these changes, ESMValTool could only be used with model output that had already been
processed to the CMOR standard such as from model intercomparison projects like CMIP. Extending ESMValTool enables the evaluation of native model output and potentially offers a simplified workflow for the CMORization process. This allows ESMValTool to be used during model development or for analysis of non-MIP-related experiments.

Software tools that allow for an easy and comprehensive evaluation of ESMs are increasingly crucial as models continue to increase in complexity and resolution. ESMValTool provides one such tool that enables comparison with observations,





reanalyses, and/or other models. The changes to ESMValTool described here are designed to lower the barrier to its use for a broad array of applications.

Along with CMOR-like data processing, ESMValTool provides regridding functionality that allows the use of flexible interpolation schemes and extends the number of available algorithms that can be used on unstructured data. In total, three schemes to interpolate unstructured grids to regular grids are now available: nearest-neighbor, bilinear, and first-order conservative regridding. While the first algorithm supports unstructured data in arbitrary format, the latter two can only be used with UGRID-compliant data. The only model that uses an unstructured grid described in this paper is ICON. Since native ICON output does not follow the UGRID standard, it can only be regridded with the nearest-neighbor algorithm in the current release of ESMValTool (v2.6.0). While this is sufficient to get a quick overview of simulation results (e.g., for monitoring of running simulations), more sophisticated schemes are needed for scientific analyses. An experimental fix to make ICON output fully UGRID-compliant during runtime has already been implemented in the ESMValTool development version and is expected to be included in future releases of ESMValTool. A number of CMIP models use unstructured grids already (e.g., E3SM, GFDL), and other models (including CESM) are likely to use unstructured grids in future versions. Global high-resolution models (e.g., participating in DYAMOND; Stevens et al., 2019) overwhelmingly use unstructured grids. Therefore, developing these regridding capabilities within ESMValTool anticipates future challenges of model evaluation and intercomparison.

The automatic CMOR-like reformatting of native model output amplifies the application of ESMValTool's wide range of diagnostics. Section 4, for example, demonstrates how ESMValTool can be used to monitor climate model simulations while they are running. For this, new diagnostics have been implemented that handle arbitrary variables from arbitrary data sets. Monitoring of running simulations facilitates the production process at modelling institutes as problems with simulations can be promptly detected. Another example is provided in Section 5, showcasing how multiple models in their native format can be easily compared with CMIP6 and reanalysis data. A further expected application of the CMOR-like reformatting is the performance assessment of new model versions or setups. For example, experiments with new parameterizations can be compared to versions of the same model with the previous parameterization scheme to assess the impact on the climate. The CMOR-like reformatting of ESMValTool can also be used simply as a CMORization of the native model output by specifying to save preprocessor output to disk. This can be particularly helpful if the model data need to be made available in CMORized form, as, for example, required by CMIP for publication of the data to the ESGF (Earth System Grid Federation) servers.

Future developments of ESMValTool will include optimizations of its parallelization capabilities and memory usage, which will allow ESMValTool to process high-resolution data provided by many modern climate models, potentially in their native format. Moreover, the implementation of the CMOR-like reformatting of native model output described in this paper is intentionally kept general and can in principle be applied to any climate model output. The five models presented here serve as examples and can be seen as a starting point for extending ESMValTool's support for native model output. As ESMValTool is a community-driven tool that is developed open-source, contributions from other modeling groups are always very welcome.



*Code availability.* The new extensions described in this paper are available since ESMValTool v2.6.0. ESMValTool v2 is released under the Apache License, VERSION 2.0. The latest release of ESMValTool v2 is publicly available on Zenodo at https://doi.org/10.5281/zenodo. 3401363 (Andela et al., 2022a). The source code of the ESMValCore package, which is installed as a dependency of ESMValTool v2, is also publicly available on Zenodo at https://doi.org/10.5281/zenodo.3387139 (Andela et al., 2022b). ESMValTool and ESMValCore are developed on the GitHub repositories available at https://github.com/ESMValGroup (last access: 1 August 2022). For further details, we refer to the ESMValTool documentation available at https://docs.esmvaltool.org/ (last access: 1 August 2022) and the ESMValTool website (https://www.esmvaltool.org/, last access: 1 August 2022).





## Appendix A: Example Extra Facets File

```
430   # File emac-mappings-example.yml

      ---

      EMAC:  # dataset name
        Amon:  # MIP table
          tas:  # CMOR variable
435         raw_name: [temp2_cav, temp2_ave]
            channel: Amon
          ta:  # defined on plev19
            raw_name: [tm1_p19_cav, tm1_p19_ave]
            channel: Amon
440     CFmon:
          ta:  # defined on hybrid levels
            raw_name: [tm1_cav, tm1_ave]
            channel: Amon
        Omon:
445       tos:
            raw_name: tsw
            channel: g3b
        '*':  # wildcards also work
          '*':
450         postproc_flag: ''
```

The YAML file above (`emac-mappings-example.yml`) showcases an example of an extra facets file. It contains small parts of the original extra facets file used to read native EMAC output. These files are project-specific, i.e., they describe extra facets for all data sets of a given project defined by the name of the extra facets file (here: *EMAC*).

Extra facets files consist of nested dictionaries with four layers. The first layer describes the name of the data set (here: 455 *EMAC*). The second and third layer correspond to the name of the MIP table (e.g., *Amon*) and the CMOR variable (e.g., *tas*), respectively. Finally, the fourth layer lists the facets that will be added to all data sets defined in the ESMValTool recipe that match the description given by the other layers. The key-value pairs given in this fourth layer are model-specific. For example, in the EMAC file given here, possible values are the raw variable name used in the EMAC netCDF files (*raw_name*), the channel name of the variable (*channel*), and a postprocessing flag that can be used to identify EMAC output files that have 460 already been postprocessed by an additional script by the modeler (*postproc_flag*). For the first three layers, wildcards are accepted, which can be used to conveniently add extra facets for multiple data sets, MIP tables, or variables at once.





## Appendix B: Application to Visualize Results of Monitoring Diagnostics

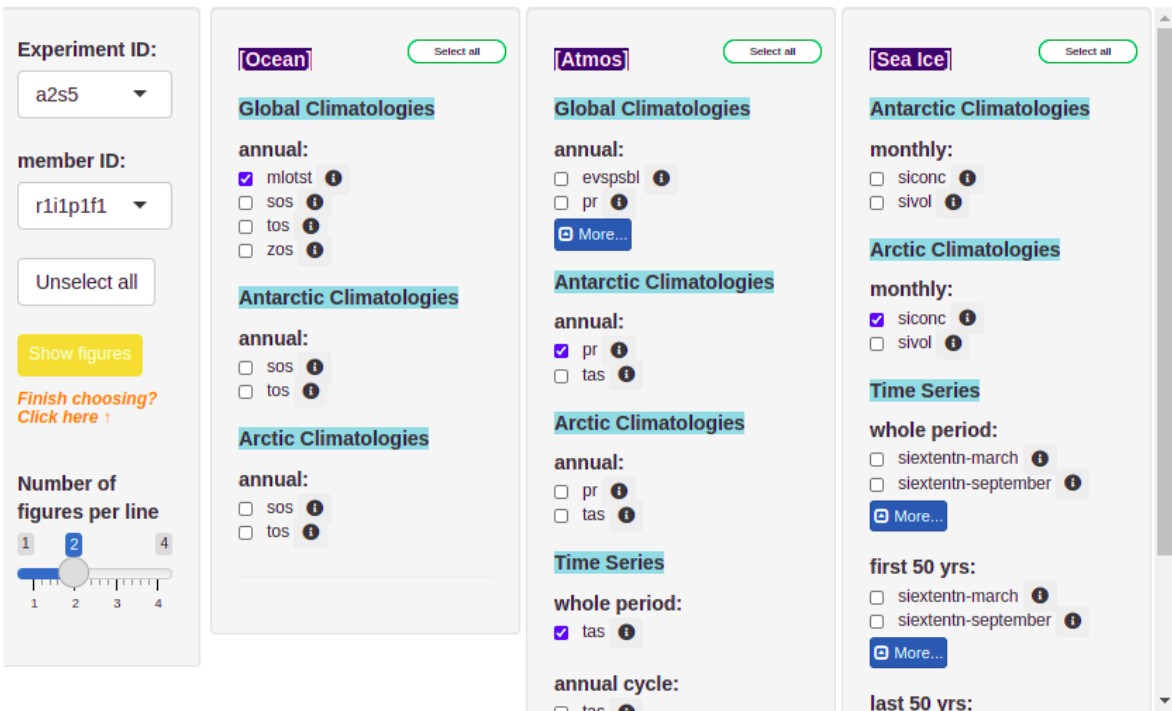

**Figure B1.** Screenshot of the R Shiny app that has been developed to conveniently and interactively visualize the results of EC-Earth3 simulation output.



*Author contributions.* MS designed the concept of this study, led the writing of the paper, implemented the CMOR-like reformatting for CESM2, EMAC and ICON, and contributed to the monitoring diagnostics. BH, AL, and VE contributed to the concept of this study. BA promoted the idea of using the fix system of ESMValTool v2 for fixing native model data. PJ provided EMAC data. SLT provided EC-Earth3 data and designed the monitoring diagnostics. BM provided CESM2 data. SS implemented the CMOR-like reformatting for IPSL-CM6 and provided IPSL-CM6 data. JS provided IPSL-CM6 data. TS provided ICON data. JV designed the fixes system of ESMValTool v2 and designed the monitoring diagnostics. KZ implemented the extended regridding functionalities presented in this study. MS, BH, AL, BA, RK, SLT, VP, SS, JV, KZ, and VE contributed to the development of ESMValTool v2. All authors contributed to the text.

*Competing interests.* Some authors are members of the editorial board of Geoscientific Model Development. The peer-review process was guided by an independent editor, and the authors have also no other competing interests to declare.

*Acknowledgements.* The development of ESMValTool is supported by several projects. This project has received funding from the European Union's Horizon 2020 research and innovation programme 4C (grant agreement No. 821003). This project has received funding from the European Union's Horizon 2020 research and innovation programme under Grant Agreement No. 101003536 (ESM2025 – Earth System Models for the Future). This project has received funding from the European Union's Horizon 2020 research and innovation programme "Infrastructure for the European Network for Earth System Modelling (IS-ENES3)" under grant agreement No. 824084. Funding for this study was provided by the European Research Council (ERC) Synergy Grant "Understanding and Modelling the Earth System with Machine Learning (USMILE)" under the Horizon 2020 research and innovation programme (Grant agreement No. 855187). BM acknowledges support by the U.S. Department of Energy under Award Number DE-SC0022070 and National Science Foundation (NSF) IA 1947282; the National Center for Atmospheric Research, which is a major facility sponsored by the NSF under Cooperative Agreement No. 1852977; and the National Oceanic and Atmospheric Administration under award NA20OAR4310392. TS acknowledges funding support from the European Research Council (ERC) under the European Union's Horizon 2020 programme (Grant agreement No. 951288). This work used resources of the Deutsches Klimarechenzentrum (DKRZ) granted by its Scientific Steering Committee (WLA) under project IDs bd0854, bd1179 and id0853. We would like to thank Mattia Righi (DLR) for helpful comments on the manuscript.





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
