# Peer review of "Evaluation of Native Earth System Model Output with ESMValTool v2.6.0"

_Geoscientific Model Development, 2022_

## Author Response (AR1)

**Answer to Anonymous Referee #1 (RC1)**

Reviewer comments are given in **bold**, our answers in blue.

**The work by Schlund and coauthors describes developments of the ESMValTool software to process non-CMOR-compliant climate model outputs, providing some examples from different models.**

**This is a practical option since as the authors write it can be used during production before ad-hoc CMORization or to process generic model outputs. Processing a certain model requires developing a set of additional files needed to perform adjustments to input files.**

We thank the reviewer for the helpful and constructive comments. We have now revised our manuscript in light of these and the other reviewer's comments we have received. A pointwise reply is given below. Line numbers in the answers correspond to line numbers in the revised manuscript.

**My major concern is related with the definition of 'native' in this context. I have the impression that input files to ESMValTool need to be already e.g. in netCDF format and CF-compliant, which is not standard for many (atmospheric) models using the GRIB format. Other model components may even provide text (i.e., tabular) outputs, which may required preprocessing. For example for EC-Earth, the authors write that the ece2cmor3 tool must be called in advance to prepare 'native' data. Please clarify this aspect.**

If supported by the native model reformatter, input files for ESMValTool do not need to be in CF-compliant netCDF format. Iris (the Python package used for file handling) can read and process netCDF and GRIB files (among other formats), and text files could be read by using the `fix_file` function. However, since all currently supported models provide netCDF output, this feature is not used at the moment. In the manuscript, we added the following sentences to clarify this aspect:

L.96–97: "*In principle, any data format for the native model output is supported (e.g., netCDF, GRIB, text files, etc.).* "

L.115–116: "*In practice, this can be useful to process native model output that is only available in unconventional file formats, e.g., in plain text files. However, this step is not necessary for the currently supported models.* "

We fully agree that the exact definition of "native" was not clearly communicated in the manuscript. Here, we use the term "native" to refer to "operational" model output, i.e., model output that is the result of the standard workflow used by the corresponding modeling center to run their model. In the manuscript, we added a clear definition of

"native" to the abstract and introduction:

L.11–12: " *[...] native climate model output, i.e., operational output produced by running the climate model through the standard workflow of the corresponding modeling institute.* "

L.65–67: " *In the context of this paper, the term "native" refers to operational output produced by running the climate model through the standard workflow of the corresponding modeling institute including potential postprocessing steps commonly used in practice.* "

For some models, this workflow may include postprocessing steps like `ece2cmor3` for EC-Earth. Thus, you are fully correct that for EC-Earth, the tool `ece2cmor3` must be run in advance; however, this is part of the default workflow. This is now also clarified in the manuscript:

L.188–192: " *As part of the standard workflow used to run the model, this data is then postprocessed to CMOR- and CF-compliant netCDF format. For this, the Python package ece2cmor3 (van den Oord, 2017, https://github.com/EC-Earth/ece2cmor3, last access: 1 November 2022) is used, which contains modules to format output from each of the model components. Thus, a CMOR-like reformatting of the native (i.e., operational) EC-Earth3 output within ESMValTool during runtime is not necessary.* "

**Are preprocessed files by ESMValTool fully CMOR-compliant or not? How is it checked?**

Yes, preprocessed files by ESMValTool are fully CMOR-compliant. This is checked with a built-in CMOR checker located here: `https://github.com/ESMValGroup/ESMValCore/blob/main/esmvalcore/cmor/check.py`. We also added a note about this in the manuscript and updated Figure 1 to reflect this:

L.94–95: " *Here, we extend the functionalities of these fixes to reformat the native model output during runtime into fully CMOR-compliant netCDF files.* "

Caption of Figure 1: " *Here, we describe additions that allow reading and processing native model output (bottom left blue ellipse) with ESMValTool through a CMOR-like reformatting (yellow round rectangles) within the ESMValCore preprocessing pipeline. As a result, the data is fully CMOR-compliant after this initial preprocessing step and can be processed by the diagnostic scripts (orange box) just like any other input data set. The diagnostic scripts do not need to treat native model output in any special way.* "

**Another important point is the availability and possibility to adapt the 'native' processing functionality. It would be important to provide clear guidelines for prospective users, and to explain how more complex calculations are carried out. A set of predefined diagnostics could be suggested for a certain (e.g. AMIP vs coupled) simulation type. Showing more examples of different diagnostics could be**

**useful too. Currently I cannot find the code you mention in the github repository, e.g. this `esmvaltool/recipes/monitor/recipe_monitor.yml` seems related but no links are given in the paper. Are there one or more recipes? Where can one find documentation in `https://docs.esmvaltool.org/en/latest/recipes/index.html`?**

We agree that the manuscript needs more guidance for potential users. For this reason, we updated several sections of the manuscript with appropriate links:

L.151–153: " *Detailed user instructions on this can be found in ESMValTool's documentation (https://docs.esmvaltool.org/en/latest/input.html#datasets-in-native-format, last access 1 November 2022). The documentation provides links with further details on all the available models and instructions on how to add support for new climate models.* "

L.330–332: " *Further details on the monitoring diagnostics can be found in ESMValTool's documentation (https://docs.esmvaltool.org/en/latest/recipes/recipe_ monitor.html, last access 1 November 2022).* "

L.445–448: " *Detailed user instructions on the CMOR-like reformatting of native model output can be found in ESMValTool's documentation at https://docs.esmvaltool.org/en/latest/ input.html#datasets-in-native-format (last access 1 November 2022). The documentation is recommended as a starting point for new users and provides links with further details on all currently supported models and instructions on how to add support for new climate models.* "

Moreover, we created an example recipe that showcases reading and processing native model output publicly available on Zenodo at `https://doi.org/10.5281/zenodo.7254312`. This recipe also provides details on the monitoring diagnostics and serves as a good starting point for potential users. We did not include this recipe into the public ESMValTool repository since the data, being preliminary model output, is not publicly available. Since data availability is a requirement for recipes integrated into the repository, we are therefore unable to integrate this modified version of the monitoring recipe into the repository. We also refer to this recipe in the manuscript:

L.333–336: " *The following paragraphs illustrate five example plots (one for each currently supported climate model) created with these new diagnostics. A recipe to reproduce these figures is publicly available on Zenodo (Schlund, 2022). This recipe showcases the usage of the monitoring diagnostics on native model output and serves as a convenient starting point for users who want to process native model output with ESMValTool.* "

L.443–445: " *An example recipe to get started with processing native model output with ESMValTool is publicly available on Zenodo at https://doi.org/10.5281/zenodo.7254312 (Schlund, 2022). This recipe reproduces Figures 2–7 of this paper.* "

In addition, we now explicitly state that the monitoring diagnostics can also be used to compare different model versions (e.g., coupled vs. uncoupled):

L.337–338: "*For a direct comparison with one or multiple reference data sets (e.g., observations, reanalyses, output from other model versions, etc.), Figure 3 shows [...].* "

Since the manuscript now gives guidelines how to get started with the evaluation of native model output and provides several examples, we do not think that showing further example diagnostics would add much value to the paper.

**Generally when using publicly available (e.g. via CMIP6), it would be better to include their identifier (e.g. r1i1p1f1) rather than generically refer to 'one realization'**

The exact name of the ensemble member has been added to the description of the corresponding figure:

Caption of Figure 6: " *The simulation shown here is an AMIP simulation that has been published as part of the CMIP6 ensemble (ensemble member r1i1p1f1).* "

**Comments by line**

**L34 model independence (Pennell 2011) should be mentioned**

Thank you for this reference. However, we think it is not necessary to mention model (in)dependence here since this particular paragraph refers to the increased complexity and degrees of freedom within *individual* models, not in the entire ensemble.

**L55 I guess Juckes 2020 should be cited**

Yes, definitely! Thank you for the reference, we added this to the manuscript.

**L62 typo 'rawoutput'**

This part has been removed from the manuscript.

**L102 I do not understand if a variable, say Tas, would support different fixes, e.g. for different frequencies or aggregations**

Yes, this would be supported. Since arbitrary code can be used in the fix class, different setups (frequencies, aggregations, etc.) can be considered. In practice, however, this has not been needed so far.

**L108 can you provide more information on these very rare cases?**

We rephrased this sentence. It now explicitly mentions a possible application:

L.113–116: " *As the very first step in the preprocessing chain, `fix_file` is meant to fix input files that cannot be read by the ESMValTool preprocessor (via the Iris module) without modifications. In practice, this can be useful to process native model output that is only available in rather unconventional file formats such as plain text files. However, this step is not necessary for the models currently supported.* "

**Fig 1 I am not sure transparency is the best choice to display additions, maybe use a specific color, shape or adding a symbol instead?**

We revised Figure 1. Now it does not contain transparency anymore.

**L119 what is the relationship between config-developer and config-user? Is a template for the former provided to all users?**

The user configuration file contains user-configurable settings like input paths, output paths, output file type, maximum number of parallel tasks, etc. A template for this file is provided with each installation of the tool, which can be obtained by running `esmvaltool config get_config_user` (see `https://github.com/ESMValGroup/ESMValCore/blob/main/esmvalcore/config-user.yml` and `https://docs.esmvaltool.org/en/latest/quickstart/configuration.html` for more information on this.

The developer configuration file contains information mostly relevant for developers, like directory structures and file naming conventions of the different types of data sets (*projects*). More information on this can be found here: `https://docs.esmvaltool.org/projects/esmvalcore/en/latest/quickstart/configure.html`.

**L162 here and elsewhere you mention that this is 'experimental' (and I think repetition could be avoided). Can you clarify why this is the case? Since output processing requires python2, does it clash with the ESMValTool environment (which I think is python3)?**

Yes, this should certainly be made clearer. We added the following description and now only mention "experimental" in the abstract and the corresponding subsection:

L.172–174: " *In contrast to the other four models presented in this paper, ESMValTool's support for native CESM2 output is still under development and thus considered experimental. Currently, only surface variables (i.e., no 3-dimensional variables with a Z-dimension) are supported.* "

Regarding Python 2: this sentence only refers the workflow that has been used to provide data for CMIP6 in the past; this code is not used in any way in ESMValTool.

**L165 I think the statement is inaccurate, see `https://ec-earth.org/consortium/`**

Thank you, this has been corrected:

L.176: " *EC-Earth3 is a global climate model developed as part of the EC-Earth consortium (Döscher et al., 2022)* "

**L171 Have you switched the sub-models?**

Yes, we did! This has been fixed in the new version of the manuscript.

**L181 So this means that ece2cmor3 can be called by ESMValTool? Or is the ece2cmor3 step required before running it?**

`ece2cmor3` is not called by ESMValTool. As mentioned in the answer to your first comment, `ece2cmor3` is part of the default workflow used to run the model. This has been clarified in the manuscript now:

L.188–192: " *As part of the standard workflow used to run the model, this data is then postprocessed to CMOR- and CF-compliant netCDF format. For this, the Python package ece2cmor3 (van den Oord, 2017, https://github.com/EC-Earth/ece2cmor3, last access: 1 November 2022) is used, which contains modules to format each of the model components. Thus, a CMOR-like reformatting of the native (i.e., operational) EC-Earth3 output within ESMValTool during runtime is not necessary.* "

**L195 'w.r.t. time' could be rephrased**

Rephrased to *temporal statistical.*

**L202 so are these 'channels' files with multiple variables/frequencies?**

Each channel contains multiple variables (*objects* in EMAC terminology) with a single frequency. However, each object may be included in multiple channels. We added three sentences to the manuscript to clarify this:

L.208–210: " *Thus, with CHANNEL, a set of model variables (called* objects*) are grouped into a* channel*. Each channel is output at a user-defined frequency as a (time-)series of files. Different channels can be output with different frequencies and objects can be part of multiple channels.* "

**L213 Does it mean that the output is CMOR-like netCDF? I can't find this in the text**

Yes, it is in netCDF format, and many variables are already in their desired (CMOR) format. We expanded this text:

L.225: " *ICON model output consists of netCDF files that already provides many CMOR variables in the correct form.* "

**L224 downloaded from where?**

From servers of the Max Planck Institute for Meteorology (MPI-M):

L.235–236: " *This grid file is specified in the global netCDF attributes of the ICON file and is automatically downloaded from MPI-M servers if necessary.* "

**L261 Add general reference on this topic for completeness?**

After a thorough search we didn't find any suitable general reference on this topic. The most relevant ones are the UGRID conventions, which are cited later in this paragraph.

**L312 Which method is used for EOFs? Diurnal, seasonal or longer-term variability should probably be subtracted to ensure EOF results are meaningful, while e.g. removal of long-term average may be insufficient**

The EOF diagnostic mentioned here simply calculates the EOFs and PCs and plots them. The preprocessing of the data is defined in the recipe by the user. An example recipe that uses this diagnostics is available here: `https://github.com/ESMValGroup/ESMValTool/blob/main/esmvaltool/recipes/monitor/recipe_monitor.yml`.

**L316 Is this app also included in ESMValTool? Can it be used for other models?**

This app is not included in ESMValTool. It is used and developed at the Barcelona Supercomputing Center (BSC) to evaluate EC-Earth output.

**Fig 3 What is 'Cool Ruby'?**

"Cool Ruby" is the name of this simulation. We rephrased this to "*called Cool Ruby*" to make this clearer in the text.

**L319 This sentence is redundant, as it can be deduced from the table already (e.g. EOFs not shown)**

We removed this sentence.

**L323 But is ERA5 CMOR-compliant? If not, how can it be used with the tool?**

Native ERA5 data is not CMOR-compliant. However, the data can be natively used by ESMValTool through a set of fixes (just like the ones for the models described in this paper). Documentation on this can be found here: `https://docs.esmvaltool.org/projects/esmvalcore/en/latest/quickstart/find_data.html#era5`. For many other observational datasets, ESMValTool provides a CMORizer script which needs to be run once before the data can be read with ESMValTool. More details on this are provided here: `https://docs.esmvaltool.org/en/latest/input.html#observations`.

**L326 I guess Gates 1992 is a more fitting reference**

Yes, indeed, we adapted the reference here.

**L373 straw comma after 'note,'. I do not know understand the iterated statements about the impossibility to compare models in this way, can you explain?**

We removed the comma. Since not all models use the same forcings, boundary conditions, and model configuration, a fair comparison of the models à la "Model X is closer to the observations than Model Y" is not possible. To avoid that readers may draw wrong conclusions from this figure we added this statement as a caveat.

**L379 It is not clear to me if an end-to-end CMORization from raw to CMOR outputs is possible for some models with ESMValTool**

This is possible for the supported models since the preprocessed output is fully CMOR-compliant and could be saved if desired by the user (see new Figure 1). For other models, we do not make any statement as this depends on the actual model output and might differ from case to case as the reviewer points out. We expanded the text here to make it clear that this feature only applies to the supported models:

L.394–395: "*The diagnostics presented in Sections 4 and 5 showcase two example applications possible with ESMValTool's new CMOR-like reformatting of native model output.*"

**Answer to Anonymous Referee #2 (RC2)**

Reviewer comments are given in **bold**, our answers in blue.

**The submitted article by Schlund et al provides details and examples on how ES-MValTool have been further developed in order to read raw output from a number of Earth System Models. The structure provided also give a framework on how to extemnd these capabilites to other models. Even though the article is very technical adding and describing these capabilities are an important step in order to use the tool efficiently, so I think it may be accepted in GMD.**

We thank the reviewer for the helpful and constructive comments. We have now revised our manuscript in light of these and the other reviewer's comments we have received. A pointwise reply is given below. Line numbers in the answers correspond to line numbers in the revised manuscript.

**The article are however a bit unclear on the first step of the process or whether there are some initial steps prior to the process not covered by the article, e.g time averaging, preprocessing, e.g. for EC-Earth. I think it might be useful if the authors provide a even more specific overview of the steps needed, i.e. in a model table. I presume 4-5 cells /categories would be enough for the purpose. E.g: Preprocessing; Create facets; Run preprocessors; ....**

We agree that the manuscript needs to be clearer on the specific steps necessary to read and process native model output. For this reason, we clarified the definition of "native" in the context of our paper: we use the term "native" to refer to "operational" model output, i.e., model output that is the result of the standard workflow used by the corresponding modeling centers to run their model:

L.11–12: "*[...] native climate model output, i.e., operational output produced by running the climate model through the standard workflow of the corresponding modeling institute.* "

L.65–67: "*In the context of this paper, the term "native" refers to operational output produced by running the climate model through the standard workflow of the corresponding modeling institute including potential postprocessing steps commonly used in practice.* "

For some models, this workflow may include postprocessing steps like applying the software package `ece2cmor3` for EC-Earth. This has been clarified in the revised manuscript:

L.188–192: "*As part of the standard workflow used to run the model, this data is then postprocessed to CMOR- and CF-compliant netCDF format. For this, the Python package ece2cmor3 (van den Oord, 2017, https://github.com/EC-Earth/ece2cmor3, last access: 1 November 2022) is used, which contains modules to format output from each of the model components. Thus, a*

*CMOR-like reformatting of the native (i.e., operational) EC-Earth3 output within ESMValTool during runtime is not necessary. "*

We also provided links to the documentation of ESMValTool that describes how to use the new feature, and how to add support for new climate models. Moreover, we added references to a publicly available ESMValTool recipe that showcases the usage of these new features:

L.151–153: " *Detailed user instructions on this can be found in ESMValTool's documentation (https://docs.esmvaltool.org/en/latest/input.html#datasets-in-native-format, last access 1 November 2022). The documentation provides links with further details on all the available models and instructions on how to add support for new climate models. "*

L.333–336: " *The following paragraphs illustrate five example plots (one for each currently supported climate model) created with these new diagnostics. A recipe to reproduce these figures is publicly available on Zenodo (Schlund, 2022). This recipe showcases the usage of the monitoring diagnostics on native model output and serves as a convenient starting point for users who want to process native model output with ESMValTool. "*

L.443–448: " *An example recipe to get started with processing native model output with ESMValTool is publicly available on Zenodo at https://doi.org/10.5281/zenodo.7254312 (Schlund, 2022). This recipe reproduces Figures 2–7 of this paper. Detailed user instructions on the CMOR-like reformatting of native model output can be found in ESMValTool's documentation at https://docs.esmvaltool.org/en/latest/input.html#datasets-in-native-format (last access 1 November 2022). The documentation is recommended as a starting point for new users and provides links with further details on all currently supported models and instructions on how to add support for new climate models. "*

We also updated Figure 1 to show the new features (and compare them to the "old" approach) in more detail.

**On a related note, while it is good that the authors show the capabilities by adding examples for all the models I also think it is a bit unclear whether this means that the process need to be different for all the models all the way until the plotting scripts. Or are there a point e.g. after the config_developer where the formats of the different model are equivalent and that later procedures/scripts can be used for all models producing the same type of plots as for the model specific example.**

**This is in particular important for section 4 and 5. While the models are clearly separated in section 2 this is not the case for section 4 and 5. Given that any plots may have been produced with any of the models this is fine. If not I think there should be a more clear separation between the models.**

After the initial preprocessing, all model output is fully CMOR-compliant. Thus,

native models do not need any special treatment for further preprocessing steps such as horizontal or vertical regridding, time averaging, area selection or masking, or the subsequent diagnostic scripts. All diagnostics (except those that require very specific input) can in principle be applied to suitable arbitrary input, as is the case for the monitoring diagnostics showed in Section 4. To make this clearer in the manuscript, we updated Figure 1 (which now shows that all native input data is fully CMOR-compliant after the preprocessing) and also expanded the text:

Caption of Figure 1: " *Here, we describe additions that allow reading and processing native model output (bottom left blue ellipse) with ESMValTool through a CMOR-like reformatting (yellow round rectangles) within the ESMValCore preprocessing pipeline. As a result, the data is fully CMOR-compliant after this initial preprocessing step and can be processed by the diagnostic scripts (orange box) just like any other input data set. The diagnostic scripts do not need to treat native model output in any special way.* "

L.380–382: " *Since preprocessed output by ESMValTool is fully CMOR-compliant for all input data sets (see Figure 1), no specific changes to these diagnostics scripts are required when dealing with native model output.* "

**Minor issues**

**Line 62. Typo: "rawoutput" –> raw output**

This part has been removed from the manuscript.

**Line 90 I presume at least some of the intermediate products be stored. Preprocessing is often heavy for long timeseries so you want to reduce the runtime. Also mentioned at line 413, but can be included already here.**

We added the following sentence to Line 90:

L.95–96: " *If desired by the user, these files can also be saved to disk, which allows ESMValTool to be used as a CMORization tool.* "

**line 238 Oceanic model –> ocean model**

Thank you, fixed.

**line 329. It is often very useful to compare the experiment with a control version. Is this easily done e.g using earlier stored fields?**

Yes, this is possible! We expanded the first sentence in the corresponding paragraph to clarify this:

L.337–338: "*For a direct comparison with one or multiple reference data sets (e.g., observations, reanalyses, output from other model versions, etc.), Figure 3 shows [...].* "

**Page 16, figure 6.  May be useful also to show if you can provide simple statistics, e.g. global averages ? bias rmse?**

Yes! We added simple statistics (global averages, bias, RMSE, and $R^2$) to Figures 5 and 6.  This feature is already available in the latest version (v2.7.0) of ESMValTool (see `https://github.com/ESMValGroup/ESMValTool/pull/2790`).

**line 403:  Any plans for developing more advanced regridding?**

Currently, there are no plans to develop regridding algorithms within the ESMValTool repository.  However, as described in the manuscript, ESMValTool now allows the usage of external regridding packages (in addition to native Iris schemes) with arbitrary options.  Thus, we can take advantage of other software packages that specialize on the development of regridding algorithms.  An example for this is the actively developed `iris-esmf-regrid` package (`https://github.com/SciTools-incubator/iris-esmf-regrid`), which allows the usage of ESMF regridding algorithms within Iris, and thus within ESMValTool.